

# Evaluating stream $CO_2$ outgassing via Drifting and Anchored flux chambers in a controlled flume experiment

Filippo Vingiani[1], Nicola Durighetto[1], Marcus Klaus[2], Jakob Schelker[3,4], Thierry Labasque[5], and Gianluca Botter[1]

[1]Department of Civil, Environmental and Architectural Engineering, University of Padua, 35131 Padua, Italy
[2]Department of Forest Ecology and Management, Swedish University of Agricultural Sciences, 901 83 Umeå, Sweden
[3]Department of Limnology and Oceanography, University of Vienna, 1090 Vienna, Austria
[4]WasserCluster Lunz GmbH, 3293 Lunz am See, Austria
[5]Géosciences Rennes, Université de Rennes 1, 35042 Rennes, France

**Correspondence:** Filippo Vingiani (filippo.vingiani@phd.unipd.it)

**Abstract.** Carbon dioxide ($CO_2$) emissions from running waters represent a key component of the global carbon cycle. However, quantifying $CO_2$ fluxes across air-water boundaries remains challenging due to practical difficulties in the estimation of reach-scale standardized gas exchange velocities ($k_{600}$) and water equilibrium concentrations. Whereas craft-made floating chambers supplied by internal $CO_2$ sensors represent a promising technique to estimate $CO_2$ fluxes from rivers, the existing

literature lacks of rigorous comparisons among differently designed chambers and deployment techniques. Moreover, as of now the uncertainty of $k_{600}$ estimates from chamber data has not been evaluated. Here, these issues were addressed analyzing the results of a flume experiment carried out in the Summer of 2019 in the Lunzer:::Rinnen - Experimental Facility (Austria). During the experiment, 100 runs were performed using two different chamber designs (namely, a Standard Chamber and a Flexible Foil chamber with an external floating system and a flexible sealing) and two different deployment modes (drifting

and anchored). The runs were performed using various combinations of discharge and channel slope, leading to variable turbulent kinetic energy dissipation rates ($2 \cdot 10^{-3} < \varepsilon < 8 \cdot 10^{-2}$ $m^2/s^3$). Estimates of gas exchange velocities were in line with the existing literature ($4 < k_{600} < 32$ m/d), with a general increase of $k_{600}$ for larger turbulent kinetic energy dissipation rates. The Flexible Foil chamber gave consistent $k_{600}$ patterns in response to changes in the slope and/or the flow rate. Moreover, Acoustic Doppler Velocimeter measurements indicated a limited increase of the turbulence induced by the Flexible Foil chamber on the

flow field (26% increase in $\varepsilon$, leading to a theoretical 6% increase in $k_{600}$). The uncertainty in the estimate of gas exchange velocities was then estimated using a Generalized Likelihood Uncertainty Estimation (GLUE) procedure. Overall, uncertainty in $k_{600}$ was moderate to high, with enhanced uncertainty in high-energy setups. For the anchored mode, the standard deviations of $k_{600}$ were between 1.6 and 8.2 m/d, whereas significantly higher values were obtained in drifting mode. Interestingly, for the Standard Chamber the uncertainty was larger (+ 20%) as compared to the Flexible Foil chamber. Our study suggests that

a Flexible Foil design and the anchored deployment might be useful techniques to enhance the robustness and the accuracy of $CO_2$ measurements in low-order streams. Furthermore, the study demonstrates the value of analytical and numerical tools in the identification of accurate estimations for gas exchange velocities. These findings have important implications for improving estimates of greenhouse gas emissions and reaeration rates in running waters.



# 1   Introduction

Growing concerns on Green House Gas emissions have increased the scientific interest in quantifying the role of inland waters in the global carbon cycle (Battin et al., 2009; Raymond et al., 2013; Hotchkiss et al., 2015; Marx et al., 2017). Most running waters are supersaturated in $CO_2$ and are believed to be responsible of a globally large biogeochemical flux to the atmosphere occurring across air-water boundaries (Horgby et al., 2019; Hall and Ulseth, 2020). In particular, first-order streams are characterized by relatively high per-area $CO_2$ evasion fluxes, and cover a significant proportion of the global stream surface (Raymond et al., 2013; Schelker et al., 2016). Therefore, accurately quantifying air-water $CO_2$ exchange in small headwater catchments is of paramount importance for global and regional assessments of $CO_2$ emissions (Rawitch et al., 2019).

According to Fick's first law of diffusion (see Eq. (1)), the flux of $CO_2$ across the air-water interface ($F$) is directly proportional to the concentration gradient between the top and the bottom of the water boundary layer through a gas exchange velocity ($k$). Thus, assessing the value of $k$, which represents the depth of the water column that equilibrates with the atmosphere per unit time, is crucial for a correct estimation of $F$. The gas exchange velocity is an intrinsic surface-water characteristic difficult to measure (Schelker et al., 2016). Moreover, $k$ can be strongly heterogeneous in space and time since it is linked to the near surface turbulence that regulates the surface renewal rate in the Mass Boundary Layer (MBL) (Zappa et al., 2003). Within reach-scale model conceptualizations, measured $k$ has been found to correlate with physical variables such as wind speed, flow velocity and channel slope (Raymond et al., 2012). These correlations originated empirical hydrogeomorphic equations that allow first-order $k$ estimation for a given reach based on simple and measurable stream attributes. Further, these equations provide means to scale $k$ across large geographic areas (Raymond et al., 2013). However, due to the inherent spatiotemporal heterogeneity of local hydraulic features, empirical laws might have limited predictability for specific case studies. In this context, independent estimates of $k$ can be useful not only to validate these empirical models (Rawitch et al., 2019) or develop more accurate scaling equations (Ulseth et al., 2019), but also to get accurate $k$ estimates for specific sites and field conditions. Such independent estimates of $k$ can be obtained through a variety of direct or indirect methods (Hall and Ulseth, 2020), including the following: i) gas tracer additions, such as propane and $SF_6$; ii) eddy covariance methods and iii) analysis of temporal dynamics of dissolved gases and iv) floating chamber measurements.

The first applications of the floating chamber method for measuring stream-atmosphere gas exchange date back to the 1960s (Department of Scientific and Industrial Research, 1964). Traditionally, the change of gas concentration inside the chamber was measured by circulating chamber air through a gas analyser (e.g. Podgrajsek et al., 2013; Gålfalk et al., 2013) or by manual sampling chamber air at distinct times and then analysing the collected samples in the lab. Over the years, the chamber's design has been continuously improved. Recently, craft-made floating chambers supplied by internal $CO_2$ sensors were proposed by Bastviken et al. (2015) as a promising technique to estimate $CO_2$ fluxes. These chambers with internal $CO_2$ sensors have since been frequently used and adapted to various applications (e.g. Lorke et al., 2015; Sawakuchi et al., 2017; Boodoo et al., 2019). The main advantages of floating chambers are their low cost, the easy-to-manage deployment, and the flexibility of the





chamber. Furthermore, chambers allow direct point measurements of gas fluxes, that are especially useful in headwater streams where flow conditions are highly heterogeneous and the interference of local $CO_2$ sources (e.g. groundwater springs) may be an issue (Ploum et al., 2018; Rawitch et al., 2019). Floating chambers can be used either in drifting mode (i.e. the chamber is

free to follow the current) or in anchored (aka stationary) mode (i.e. the chamber is fixed on a suitable support that prevents drifting). In running waters, floating chambers are preferably employed in the drifting mode (Alin et al., 2011; Beaulieu et al., 2012; Lorke et al., 2015), because anchored chambers can induce turbulence and enhance observed gas exchange rates across the water-atmosphere interface (Lorke et al., 2015). The applicability of the drifting deployment, however, is confined to low surface-roughness flow systems, where the chamber is allowed to move downstream while maintaining the necessary stability.

Despite the recent spread of the use of chambers for quantifying $CO_2$ fluxes from inland waters, the literature lacks of rigorous comparisons among differently designed chambers and deployment techniques to address the problem of $k$ misestimation due to chamber-induced turbulence. To date, only few comparative studies exist (e.g. Vachon et al., 2010; Lorke et al., 2015), and all of them highlighted the need to better clarify the impact of chamber design on the reliability of $k$ estimates. Furthermore, the uncertainty associated with the estimates of $k$ from $CO_2$ concentrations derived using chambers has not been

discussed nor quantified by previous studies. In the literature, the association of chamber-based estimates of $k$ to specific flow conditions relies on simple averages among replicate experiments, and the inherent variability of $k$ due to poor model structures or heterogeneous deployment modes is not accounted for. As errors in $k$ propagate directly in the corresponding flux estimates, uncertainty in the estimates of gas exchange velocity can strongly limit our ability to investigate in-stream biogeochemical processes that rely on air-water $CO_2$ exchange.

We pose the following two main research questions for this study: 1) is it possible to improve the reliability of $k$ measurement by $CO_2$ chambers through a novel chamber design?; 2) can we define a robust procedure to interpret the data derived from chamber experiments accounting for parameters uncertainty? Specifically, we focus on the usage of chamber methodology in an effort to i) comparatively analyse the impact of chamber design improvements suggested by the recent literature (e.g. Sand-Jensen and Staehr, 2012; Lorke et al., 2015) and ii) provide guidelines to interpret $k$ values derived from chambers. With

identifying novel procedures to analyse chamber data, we aim to offer useful tips to improve the robustness and reliability of chamber-based $CO_2$ measurements in first-order streams.

## 2  Methods

### 2.1  Instrument

In this study, we used two different chamber designs, represented in Fig. 1. While both the chambers utilize the same internal

$CO_2$ logger, they differ in the sealing design. The chamber shown in Fig. 1a (hereafter "Standard chamber") is a traditional chamber with a rigid floating wall that guarantees the isolation of the air inside the chamber (Bastviken et al., 2015). The chamber shown in Fig. 1b (hereafter "Flexible Foil chamber"), instead, contains an external floating system and is equipped with a thin flexible sealing at the water surface. The polyethylene foil (Plastic sheet UV4, Foliarex, Poznań, Poland) had a thickness of 150 μm and an height of 4 cm. The foil was mounted on the lower internal profile of the chamber cup. An





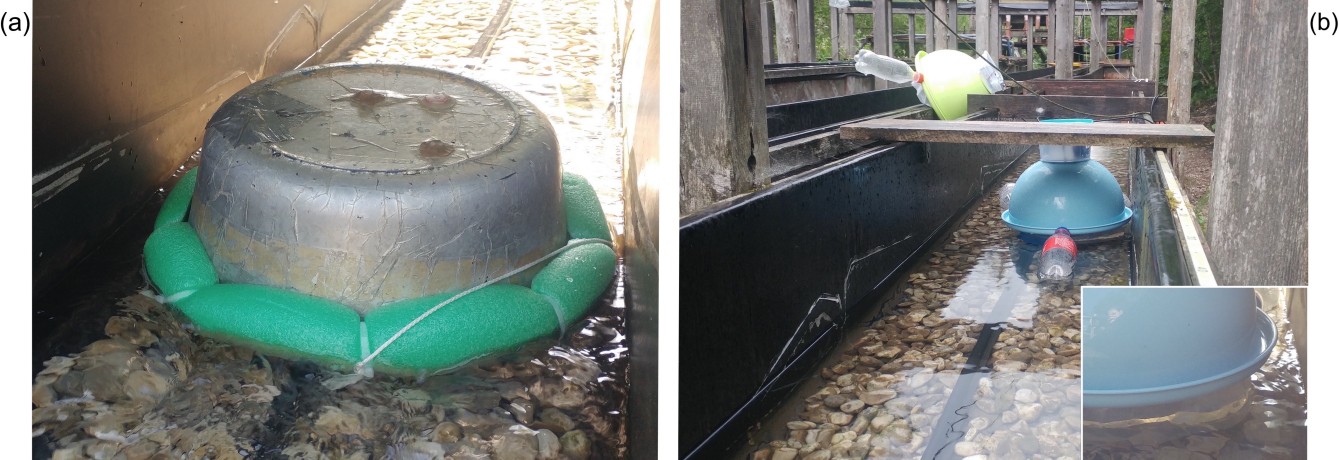

**Figure 1.** Two chambers design: on panel (a) the Standard chamber from Bastviken et al. (2015) instead on panel (b) the Flexible Foil chamber (with a close-up of the peculiarity of the sealing).

**Table 1.** Summary of the characteristics and geometrical properties of the two chambers.

| Chamber design | Material | Floating device | Area [m$^2$] | Volume [m$^3$] |
|---|---|---|---|---|
| Flexible foil (this study) | Plastic hat with flexible foil sealing | Plastic bottles | 0.0471 | 0.0054 |
| Standard (Bastviken et al., 2015) | Plastic hat cup | Styrofoam | 0.0855 | 0.00625 |

adhesive tape (Extra Power Universal, Tesa, Hamburg, Germany) was used to fix the foil to the chamber and to guarantee the isolation of the air inside the chamber from the external atmosphere. The floating system (see Appendix C) was designed via three half-liter water bottles that were fixed on the external cup margin to sustain the rigid cup of the chamber above the water surface by about 2 cm (corresponding to half of the flexible foil height). The sealing was developed to reduce the turbulence induced by the chamber, especially when used in the anchored mode, as suggested by Lorke et al. (2015). A flexible foil sealing

had been already utilized by Sand-Jensen and Staehr (2012). However, the sealing used by Sand-Jensen and Staehr (2012) was not complete, as it was implemented only in the direction of the water flow. A complete flexible sealing is particularly useful for drifting chambers that may rotate during drifting, and in turbulent flow fields where the dominant flow direction might be heterogeneous in space and time. Other geometrical characteristics of the two chambers are detailed in Table 1. The volume of the Flexible Foil chamber was computed taking into account the floating attitude of the chamber and the actual fraction of

sealing foil below the channel surface.



**Table 2.** Summary of the configurations setup used at the EcoCatch Flumes of the Lunz Mesocosm Infrastructure.

| Configuration | Discharge [$\mathrm{l\,s^{-1}}$] | Flow velocity [$\mathrm{m\,s^{-1}}$] | Travel Time [s] | Slope [%] |
|---|---|---|---|---|
| 1 | 2.74 | 0.083 | 421 | 0.05 |
| 2 | 5.50 | 0.126 | 278 | 0.05 |
| 3 | 5.63 | 0.202 | 173 | 0.25 |
| 4 | 7.04 | 0.261 | 134 | 0.25 |

The sensor (K33 ELG, SenseAir, Delsbo, Sweden) detects $CO_2$ by non-dispersive infrared (NDIR) spectroscopy up to 10000 ppm with a precision of $\pm$ 3 % of the measured value. Details on the sensor and its performance are given by Bastviken et al. (2015).

## 2.2 Study setup

We performed a total of 100 experiments in the Lunzer:::Rinnen - Experimental Flumes in Lunz am See, Austria. The flumes are fed by the Oberer Seebach and thus they are representative of a natural headwater stream. The flumes enabled the realization of a series of experiments without lateral input, in which important hydrogeomophological factors (such as flow, velocity and slope) could be controlled. Measured data (air temperature and $CO_2$ concentrations in the air and within the chamber) were collected with a variable frequency during each experiment's day depending on environmental and technical constraints. We

performed a total of 55 drifting and 45 anchored chamber measurements. The data were collected using two different flumes at the Mesocosm facility. Both flumes were characterized by a gravel bed, and individual gravels had a typical max and min axes lengths of 33 and 13 mm. Each flume was characterised by a distinct slope, and was run with two different discharge configurations (Table 2). Discharge was quantified using three different methods: salt injection, bucket, and constant rate injection. For each experimental setup, we carried out 1-5 slug injections, 5-10 bucket measurements, and 2-12 constant rate

injections. For the bucket method, we tracked the time needed to fill a 12.8 l bucket. For the slug and constant rate injection, we followed the protocols suggested by Moore (2004, 2005). In particular, we added a diluted salt tracer (NaCl+Water) at the flume inlet, and monitored electrical conductivity at intervals of 1 s (slug injection) or 30 s (constant rate injection) using loggers (HOBO U24, Onset, Bourne, MA, U.S.A.) placed 5 m downstream of the injection point and at the outlet. Constant rate injection was achieved by a peristaltic pump (Schlauchpumpe MV-GE, Ismatec, Wertheim, Germany). Next, we also measured

the travel time in the flume system, calculated as the interval between the times when 50% of the salt had passed the inlet and outlet loggers. Last, we computed the flow velocity ($v$) as the ratio between the distance between the inlet and outlet loggers (35 m) and the travel time.

Elaborating the $CO_2$ observations required the knowledge of the air atmospheric pressure; such data were derived from the meteorological station Lunz am See, located 300m from the flumes.





### 2.3 Sampling description

The sensors installed within the chambers recorded the time variability of $CO_2$ concentration in the chamber volume during a run. The duration of each run was dependent on the underlying hydraulic conditions and the deployment mode. In the drifting mode, measurements were taken for a pre-defined maximum duration, that corresponds to the travel time of the chamber in the flume. In the anchored mode the measurements lasted until near equilibrium conditions were reached.

Time series of $CO_2$ during each run were obtained with a measurement frequency of 30 s. To avoid biases in the saturation curves, it was important to aerate the chamber volume before placing the chamber above the stream and make a new run. It was observed that leaving the chamber aerated for five minutes prior to each run allowed the equilibrium between the chamber and the atmosphere to be reached. Accordingly, the selected procedure was to leave each chamber aerated for 10 minutes (2 times the estimated equilibrium time) before it was placed on the stream to make a new run. Despite this precautions, some of the measurements had to be discarded because the $CO_2$ concentration in the chamber was not in equilibrium with the atmospheric value at the beginning of the run. Regarding the calibration procedure in the field, we performed a post-process calibration referring to a constant $CO_2$ concentration in the atmosphere equal to the reference value of 400 ppm for the entire experiment. This was necessary at times to eliminate long-term drifts of the sensors induced by the high humidity in the chamber air. Atmospheric $CO_2$ concentrations were also monitored during the experiment with a sensor placed in the open air 3 m above the ground close to the flumes.

### 2.4 Chamber based estimates of mass transfer at the water-air interface

The gas flux across atmosphere-water interfaces ($F$) is commonly evaluated with the Fick's first law of diffusion (Wanninkhof et al., 2009):

$$F = k\left(C_e - C_0\right) \tag{1}$$

where $F$ is the $CO_2$ flux, [$\mu$mol s$^{-1}$ m$^{-2}$]; $k$ the gas exchange velocity, [m s$^{-1}$]; $C_e$ the concentration in the water, [$\mu$mol m$^{-3}$]; $C_0$ the concentration in the water as if it was in equilibrium with the atmosphere, [$\mu$mol m$^{-3}$]. Henry's Law was used to get the water equilibrium concentration, $C_e$ in Eq. (1), expressed as a function of the air concentration; the expression from Sander (2015) was employed to account for the dependence of Henry's constant ($K_H$) on air temperature. Eq. (1) shows that $k$ is a key parameter that governs the mass exchange across water-air interfaces. Two methods are available to estimate $k$ from chamber measurements. Both methods are based on the definition of $F$ at the air-water interface as the variation of the gas concentration inside the chamber per unit of time and area:

$$F = \frac{dC'}{dt}\frac{n}{A}. \tag{2}$$

In Eq. (2), $C'$ is the concentration measured inside the floating chamber in [ppm =$\mu$mol mol$^{-1}$]; $A$ is the exchange surface in [m$^2$] and $n$ is the number of moles in the air volume inside the floating chamber as per the gas law.

According to Eq. (2), the flux can be computed from the derivative of the chamber concentration in time. The first method (hereafter Method 1) is based on the idea that if we select a sufficiently small interval (during which the exchange process





is not influenced by the gas chamber concentration), the increments of concentration inside the chamber itself are linearly proportional to time; therefore, the flux remains constant and equal to the slope of the line interpolating the observed $CO_2$ concentrations over time. In particular, the velocity $k$ in [$ms^{-1}$] can be computed as the ratio between the measured flux (Eq.

(2)) and the difference between the gas concentration in water, $C_e$, and the gas concentration in water as if it was in equilibrium with the atmosphere, $C_0$:

$$k = \frac{dC'/dt\,n}{A}\frac{1}{C_e - C_0},\tag{3}$$

Observations are fitted via a linear model and then the slope (i.e. $dC'/dt$) is used to estimate $k$. The application of Method 1 relies on the knowledge of the value of $C_e$, which is usually obtained from either direct or indirect measurements. However,

$C_e$ can be obtained also from the $CO_2$ saturation curve provided by a steady floating chamber, i.e. from the saturation curve of a chamber allowed to reach near-equilibrium conditions as in Bastviken et al. (2015). Our estimates of $k$ from the chamber employed with drifting method use the $C_e$ obtained from the steady chambers avoiding biases associated to the calibration of different instruments. To this aim, the mean $C_e$ observed by simultaneously deployed steady chambers was used. During the experiment, we got at least one $C_e$ measure for each drifting run in the range of one hour, that was used for the interpretation

of the chambers data. Peter et al. (2014) showed that the natural diel variability of $CO_2$ concentration in the stream feeding the flume is on average $375 \pm 133$ μatm per day in summer. In this work, a maximum variation of 100 ppm within one hour for $CO_2$ in stream water would result. Therefore, the estimates of $k$ could be altered by 1.2 % at most.

An alternative method (Method 2) is proposed here to interpret the results from steady deployments. According to Eq. (2), $F$ can be also estimated as the derivative of the concentration's curve in time up to the achievement of near-equilibrium conditions

between the water and the chamber's air. In this case, the decrease of $F$ during the run is explicitly accounted for. The analytical solution of the differential equation that links the $CO_2$ concentration inside the chamber and the $CO_2$ concentration in the water (similar to Bastviken et al. (2004)) is the following:

$$C'(t) = C'_e - (C'_e - C'_0)\,exp\left[-k\,(t - t_0)\,\frac{A\,p}{n\,K_H}\right],\tag{4}$$

where $C'_0 = C'(t = t_0)$ is the concentration in the chamber in [ppm] at time $t_0$ ($C'_0 \simeq 400$ ppm in this study); and $C'_e$ the

concentration of $CO_2$ in [ppm] in the chamber air as if the air inside the chamber was in equilibrium with water. $k$ and $C'_e$ are estimated by fitting (using a minimum least square method) the exponential curve of Eq. (4) to the time series of $CO_2$ observations in the chamber during a steady run. As for Method 1, the assumption is that during the measurement the concentration of $CO_2$ in the water below the chamber remains constant. Therefore, this assumption will gain more reliable results for short saturation curves. Given the maximum observed diel cycle of $CO_2$ in the stream feeding the flumes (Peter

et al., 2014), our assumption can be considered reasonable.

The main advantages of this method with respect to Method 1 are twofold: i) independent measures of $C_e$ are not needed, ii) the estimate of the gas exchange velocity is much more robust because it is calibrated over a larger set of observations. This method is best applied to cases in which a robust estimate of the equilibrium concentration is possible, e.g. because the full saturation curve is available, as for our steady deployments. Drifting deployments were analysed with Method 1, whereas

anchored deployments were analysed via Method 2.





Regardless of the method used for $k$ estimates, chamber concentration data were checked for quality after the experiment and some runs were discarded before the analysis. The retained runs fulfil the following requirements:

- $CO_2$ curves display an increasing monotonous trend during the whole run.

- Saturation concentration in water must fall within a pre-defined reasonable range, 400 to 2000 ppm (e.g. Peter et al., 2014).

- Nash-Sutcliffe Efficiency (NSE) coefficients resulting from the model should be higher than 0.98, to avoid misestimate of $k$.

- There should be at least 2 minutes of constant $CO_2$ measurements prior to the beginning of each saturation curve; this ensures the chamber was in equilibrium with the atmosphere before the saturation curve was taken.

Before applying the check for data quality, some runs were discarded a priori because of missing data caused by technical problems with the chambers such as low battery voltage or accidental cables' disconnections.

The analysis was carried out by standardizing $k$ to a Schmidt number ($Sc$) of 600 ($k_{600}$) via the commonly used expression for $Sc_{CO2}$ (see Appendix B). A total of 40 estimates of $k_{600}$ were obtained with anchored (20) and drifting (20) deployments. The number of $k_{600}$ estimates gathered through the Standard chamber (28) largely exceeded the number of estimates from the Flexible Foil chamber (12); this was a result of the fact that three Standard chambers but only one Flexible Foil chamber were used in the experiment.

### 2.5 Comparison with existing scaling laws

Our data were used to compare the observed dependence of $k$ on the energy dissipation rate with two scaling relationships widely accepted in the literature. The first scaling law (SL1) was introduced by Zappa et al. (2003) whereas the second (SL2) was proposed by Ulseth et al. (2019). The relevant scaling laws are described in the following paragraphs.

According to the surface renewal theory, a continuous random renewal of the aqueous Mass Boundary Layer with the bulk water below is observed due to turbulent eddies. Because the MBL is a thin layer at the air-water interface that govern the transport, the gas exchange is in turn influenced by hydrodynamic flow characteristics. This is quantified through the scaling equation SL1 as:

$$k_{600} = \alpha(\varepsilon\nu)^{1/4}600^{-1/2}86400,$$
(5)

where 86400 is a conversion factor from s to days, $\nu$ is the kinematic viscosity, $\alpha$ is an empirical coefficient and $\varepsilon$ is the turbulent kinetic energy (TKE) dissipation rate that was derived from the measurements of an Acoustic Doppler Velocity (ADV) meter (Vectrino+, downlooking probe, Nortek, Rud, Norway - see Appendix A). SL1 links $k_{600}$ and $\varepsilon$ via a power law relationship with an exponent of $1/4$. We calibrated $\alpha$ via the least squared method based on the data derived from the deployment of the Flexible Foil and Standard chambers via the steady method. Then the resulting values of $\alpha$ were compared with the range 0.16 to 0.43 observed in previous studies (Moog and Jirka, 1999; Zappa et al., 2007; Tokoro et al., 2008; Vachon et al., 2010).





The second scaling law used in this study (SL2) linearly correlates $\log(k_{600})$ and $\log(\varepsilon_{emp})$, where $\varepsilon_{emp}$ is the turbulent kinetic energy dissipation rate computed as the product between channel slope, flow velocity and gravitational acceleration. SL2 was empirically derived from the analysis of a large number of gas exchange velocities estimated from tracer gas additions over a wide range of streams worldwide (Ulseth et al., 2019). The analytical expression that links $k_{600}$ and $\epsilon_{emp}$ is analogous to Eq. (5), but in this case the values of the slope and the intercept of the scaling law are 0.35 and 3.1, respectively.

### 2.6 Uncertainty analysis

The uncertainty in the estimate of gas-exchange velocity and equilibrium concentration via the exponential model was analysed using an informal Monte Carlo technique (i.e. Generalized Likelihood Uncertainty Estimation (GLUE), Beven and Binley (1992)). A random sample (one million) of couples ($k$ and $C'_e$) was generated from a bi-variate, bounded, uniform distribution and then the goodness of the exponential fitting was evaluated for each combination of parameters by means of the NSE. The range for NSE is $(-\infty, 1]$: NSE = 1 corresponds to a perfect match the model to the observed data and NSE = 0 indicates that the model predictions are only as accurate as the mean of the observed data. Afterwards, only the couples of parameters providing satisfactory performance (i.e. NSE above a threhsold value) were retained to derive the posterior bi-variate probability distribution of $k$ and $C'_e$. This technique is widely employed in the analysis of the uncertainty of the parameters of hydrological models, but it has not been used previously in the context of $CO_2$ studies. The method can be used to asses the uncertainty associated with the identification of model parameters via the fitting procedure of Method 1 and 2, regardless of the chamber type (see Sect. 3.6).

## 3 Results and Discussion

### 3.1 Summary of water concentration and gas exchange velocity estimations

Stream $CO_2$ concentrations were estimated based on $CO_2$ data gathered through steady chambers (see Sect. 2.4, Method 2). Estimated $C'_e$ values during the whole experiment were in the range 555 to 1057 ppm, with a mean of 775 ppm and a standard deviation of 166 ppm. Regardless of the chamber type, measurements of $C'_e$ from simultaneously deployed chambers exhibited a moderate variability; the mean coefficient of variation ($cv$) across different measurements, $\bar{cv}$, was around 0.15. All measurements showed supersaturation of $CO_2$ with respect to the atmosphere, thereby implying that during the deployment of the chambers the sensors recorded an increase of $CO_2$ concentration over time (i.e. a positive flux from the stream to the atmosphere). General daily patterns of $CO_2$ in the stream (Peter et al., 2014) were difficult to identify because of the limited chamber deployments frequency.

Overall, four different setups were analyzed. Each setup was characterised by a different combination of channel slope/discharge, thereby leading to a different value of $\varepsilon_{emp}$. Mean values ($\mu$) and coefficient of variation ($cv$) of $k_{600}$ for different configurations (i.e. channel slope/discharge combinations) are displayed in Table 3 - the subscripts "FF" and "Std" indicate Flexible Foil and Standard chamber respectively. Overall, mean $k_{600}$ for the four discharge/channel slope combinations ranged between



4.0 and 31.3 $\mathrm{m\,d^{-1}}$. To analyse the differences in $k_{600}$ values from the Standard and Flexible Foil chambers, we performed

a two-sample t-test assuming equal variances for the two groups (varStd and varFF) in accordance with Bonett's test (H0:

varStd/varFF=1, p = 0.79). The two-sample t-test showed a higher mean $k_{600}$ from the Flexible Foil chamber with respect to

the Standard chamber (H0: $k_{600,FF} > k_{600,Std}$, p > 0.97), particularly for high channel slopes. Moreover, $k_{600}$ derived from

Flexible Foil chambers had a smaller variability among replicated runs in the same configuration ($\bar{cv} = 0.24$) with respect to

the Standard chambers ($\bar{cv} = 0.55$), even in cases where the number of replicates was similar. Gas exchange velocities derived

from drifting and steady analysis showed a similar variability across the different setups, with a range of 4.7 to 31.3 $\mathrm{m\,d^{-1}}$

and 4.0 to 27.8 $\mathrm{m\,d^{-1}}$ for the anchored and the drifting chamber, respectively. The range of values of $\varepsilon_{emp}$ tested during the

experiment covered more than one order of magnitude (from about $2.5 \cdot 10^{-3}$ to $8.4 \cdot 10^{-2}$ $\mathrm{m^2/s^3}$). Overall, the corresponding

estimates of $k_{600}$ were in line with previous studies available from the literature for similar values of turbulent kinetic energy

dissipation rates (Raymond et al., 2012; Schelker et al., 2016; Maurice et al., 2017). The impact of relevant physical properties

of the flume, chamber type and deployment mode on $k_{600}$ are discussed in the following sections.

## 265 3.2 Dependence of gas exchange velocity on slope and flow velocity

Least-squared linear regression analysis of the observations obtained using the Flexible Foil chambers in steady mode suggested

that the gas-exchange velocity increased with flow velocity ($R^2 = 0.84$, n = 7, p = 0.004), in line with previous results (e.g.

Raymond et al., 2012; Hall and Ulseth, 2020). In contrast, the Standard chamber in anchored mode showed no significant

relationship between flow velocity and $k_{600}$ ($R^2 = 0.17$, n = 13, p = 0.168). Instead, the interpretation of data from the drifting

deployments was less straightforward. Data from the Flexible Foil chamber confirmed the trend observed in steady mode, even

though the value of $k_{600}$ relative to setup 1 was unfortunately not measured (Table 3). In contrast, drifting data from Standard

chamber showed a contrasting trend between $k_{600}$ and $v$. In particular, $k_{600}$ appeared to be unaffected by $v$ for low channel

slopes (setup 1 vs. setup 2), and inversely related to flow velocity for high channel slopes (setup 3 vs. setup 4).

The application of a least-square linear regression model to our observations from Flexible Foil chamber, both under drifting

and anchored mode, indicated a positive relationship between $k_{600}$ and channel slope (drifting: $R^2 = 0.90$, n = 5, p = 0.012;

anchored: $R^2 = 0.73$, n = 7, p = 0.015), in line with previous studies (e.g. Raymond et al., 2012; Lorke et al., 2019; Hall and

Ulseth, 2020). On the contrary, the gas exchange velocities measured via the Standard chamber did not provide a positive

correlation between $k_{600}$ and the channel slope neither in drifting nor in anchored mode.

In conclusion, for the Standard Chamber we observed unexpected patterns of $k_{600}$ for changing channel slopes and velocities,

which were not in line with previous literature results. This might be an indication that the measurements taken using the

Flexible Foil chamber were more reliable during our experiment.

### 3.3 Chamber's deployment: Anchored vs. Drifting

At the flume facility, the drifting deployment was more complicated to run with respect to the steady deployment because of

the high probability that the chamber - free to follow the current - bumped into the flumes sidewalls or interfered with the

285 gravel bed. Despite the inherent difficulties associated to the estimation of $k_{600}$ with the drifting methods, the Flexible Foil



**Table 3.** Summary of $\varepsilon$, the mean and coefficients of variation of the standardised gas exchange velocities, $k_{600}$, observed during the different configurations in anchored and drifting deployments both for the Standard (Std) and Flexible Foil (FF) chamber. The number between brackets in the column of the mean value indicates the number of measurements.

| Configuration | $\varepsilon$ $[\mathrm{m^2\,s^{-3}}]$ | mode | $\mu k_{600,Std}$ $[\mathrm{m\,d^{-1}}]$ | $cv k_{600,Std}$ $[/]$ | $\mu k_{600,FF}$ $[\mathrm{m\,d^{-1}}]$ | $cv k_{600,FF}$ $[/]$ |
|---|---|---|---|---|---|---|
| 1 | $2.44 \cdot 10^{-03}$ | steady | 5.5 (3) | 0.33 | 4.7 (1) | - |
|   |   | drifting | 4.0 (3) | 0.74 | - | - |
| 2 | $3.91 \cdot 10^{-03}$ | steady | 9.8 (2) | 0.59 | 10.8 (3) | 0.30 |
|   |   | drifting | 5.1 (2) | 0.17 | 6.8 (2) | 0.17 |
| 3 | $4.67 \cdot 10^{-02}$ | steady | 9.8 (5) | 0.64 | 21.5 (1) | - |
|   |   | drifting | 20.1 (6) | 0.76 | 21.9 (2) | 0.15 |
| 4 | $8.39 \cdot 10^{-02}$ | steady | 13.2 (3) | 0.71 | 31.3 (2) | 0.35 |
|   |   | drifting | 8.2 (4) | 0.47 | 27.8 (1) | - |

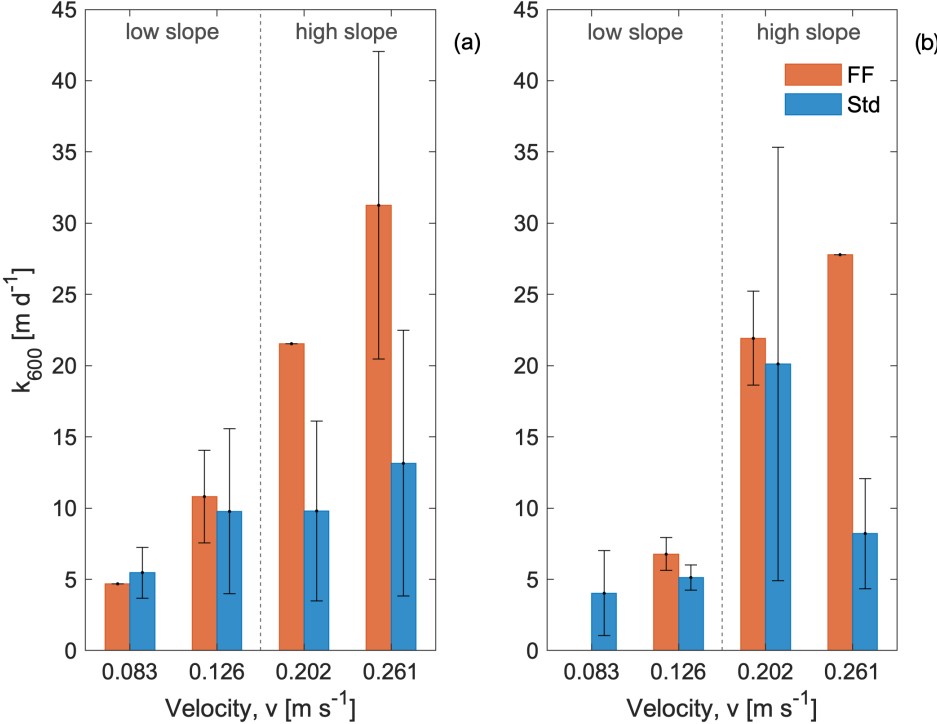

**Figure 2.** Barplot of the $k_{600}$ values as a function of channel slope and flow velocity for steady (panel (a)) and drifting (panel (b)) modes. Error bars (standard deviations) are inserted to visualize the variability among replicate measurements.




chamber gave consistent results between drifting and anchored modes for almost all the combinations (Table 3 and Fig. 2). On the contrary, the Standard chamber gave different (up to a factor of 2) estimates of $k_{600}$ in the two modes (anchored vs. drifting). This is possibly explained by the enhanced influence of external forces and/or hurdles on the Standard chamber, in which the isolation wall is not flexible and, thus, more prone to external interference. Therefore, we propose that the observed differences between drifting and steady deployments for the Standard chamber were likely caused by the chamber design.

These results indicate that the Flexible Foil chamber might be more adaptable to diverse field conditions than the Standard chamber, especially in case of shallow water flow or in streams with complex bank morphometry which makes chamber collisions more likely. In particular, we hypothesize that the physical separation between the chamber and the floating system could reduce the direct interference of hurdles in the flow field on the water surface underlying the chamber volume. However, further tests would be needed to confirm this hypothesis.

The estimate of $k_{600}$ from drifting chambers was performed here assuming that the equilibrium concentration was known (and equal to the values derived from the steady chambers, see Sect. 2.4 and 3.1). Nevertheless, even if we assume that $C'_e$ is known, $k_{600}$ estimations derived from drifting deployments might still have a limited reliability. In fact, given the limited duration of our drifting experiments (see Sect. 2.4), only few $CO_2$ observations were available for most runs, with small concentration differences among the obtained measurements. Consequently, in this setting the limited accuracy of gas concentration measurements ($\pm$ 3 % of the measured value) can strongly bias the estimate of the slope $dC'/dt$ in Eq. (2) (Mannich et al., 2019). Also, in cases where $C'_e$ were unknown, the uncertainty of the estimation of $k_{600}$ would further increase (see Sect. 3.6). Therefore, we conclude that measuring equilibrium $CO_2$ concentrations in the water during $k_{600}$ measurements is a significant advantage of steady deployments because joint estimates of $k_{600}$ and $C'_e$ are allowed based on observed $CO_2$ saturation curves in the chamber.

### 3.4 Gas exchange velocity and turbulent kinetic energy dissipation rate: Standard vs. Flexible Foil chamber

The gas exchange velocities estimated through the Flexible Foil chamber are generally higher that those estimated with the Standard chamber especially for high channel slopes (see Sect. 3.1). A tentative explanation could be that the Flexible Foil chamber, despite its design, enhances $\varepsilon$ characteristic of the flow more than the Standard chamber. To test this hypothesis, ADV measurements were used (depth of measuring volume set to 3.5 cm) to determine $\varepsilon$, associated to the undisturbed flow and to the flows below the chambers during steady runs. The comparison of $\varepsilon$ indicated a 26 % and a 84 % increase of $\varepsilon$ with respect to undisturbed flow in the case of setup 3 for the Flexible Foil and the Standard chamber, respectively. The ADV measurements showed that the rigid wall of the Standard design enhanced the turbulence on the exchange surface more than the Flexible Foil. The application of Eq. (5) to our ADV measurements indicated that the turbulence induced by the Flexible Foil chamber yields a slight bias in the estimate of $k_{600}$ (+ 6%). Instead, the Standard chamber enhanced the flow turbulence to a larger extent, causing a substantial overestimation of $k_{600}$ (+18 %). The limited observed increase of the chamber-induced turbulence in the flow field makes the Flexible Foil chamber a promising tool to estimate gas exchange rates in low order streams.





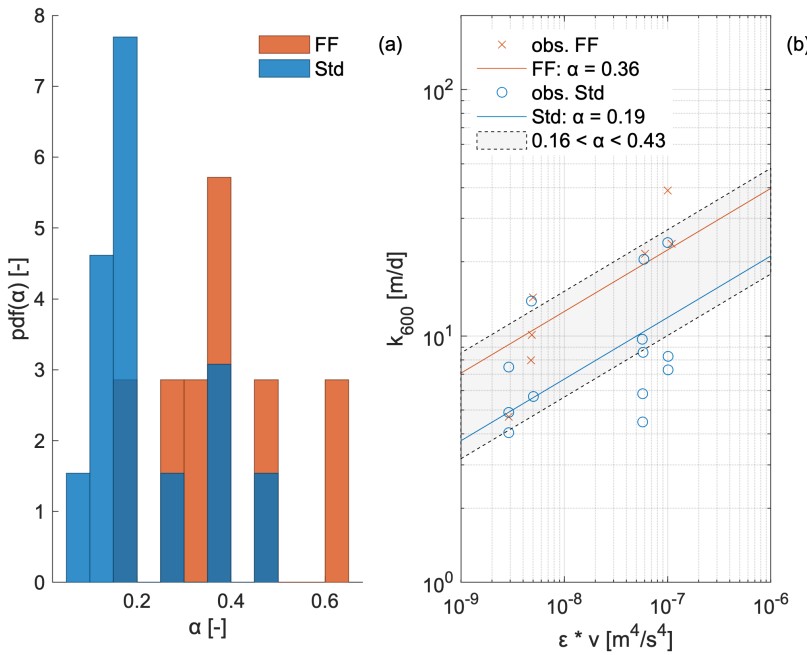

**Figure 3.** Panel (a) contains the probability density function (pdf) of $\alpha$ coefficient calibrated for each saturation curve for both the Flexible Foil and Standard chamber. Panel (b) shows Eq. 5 obtained for both the Standard and the Flexible Foil chamber in a log-log plot; coefficient $\alpha$ was calibrated based on the two data sets using the least squared method.

Another possible explanation of the lower values of $k_{600}$ in the Standard chamber, despite the higher turbulence observed

below the chamber, could be a double diffusion process that took place in the Standard chamber during the runs. In fact, the $CO_2$ sensor of the Standard chamber, differently from the Flexible Foil chamber, was protected by a perforated cover that could have slowed down the $CO_2$ flux from the water surface to the box where the sensor was placed. Accordingly, the holed cover could have induced a delay in the $CO_2$ response of the sensor, thereby reducing the observed values of $k_{600}$. While the above explanation is intriguing, there are no direct evidence for supporting or rejecting this hypothesis, and the issue is left to further

studies.

### 3.5   Comparison with existing scaling laws

Both Flexible Foil and Standard chambers in anchored mode showed increasing values of $k_{600}$ when the energy dissipation rate of the flow increased. However, the increase in gas exchange velocities over a unit of energy dissipation rate was markedly different for the two chambers under anchored mode. The analysis of data gathered in a drifting mode did not show the

expected pattern of $k_{600}$ with $\varepsilon$ for the Standard chamber. Instead, the Flexible Foil chamber produced consistent results between anchored and drifting deployments, with a monotonous increase of $k_{600}$ with $\varepsilon$ (Fig. 3b).



When we compared the relationship of $k_{600}$ with $\varepsilon$ of our measured data with the scaling relationship SL1, we observed both chambers giving reasonable results (see Fig. 3b). In Fig. 3a we show the overall probability density function (pdf) of the coefficient $\alpha$ in Eq. (5) calibrated for each single saturation curve for Flexible Foil and Standard chambers across all the runs.

Calibrating $\alpha$ on the single saturation curve allowed the variability of $\alpha$ throughout the experiment to be determined for both the chambers. In Fig. 3b we show the scaling equations (orange and blue) we got through the calibration of $\alpha$ based on all the measurements. The result further enforces the robustness of the data from our steady chambers, since both the curves fall inside the literature range (represented as a light grey shaded area). Our results showed a significant linear correlation between $\log(\varepsilon_{emp})$ and $\log(k_{600})$ ($R^2$ = 0.23, n = 20, p = 0.033), in line with Ulseth et al. (2019). By performing a least-squared

regression on the data gathered using the chambers we noted that only the Flexible Foil chamber identified a statistically significant positive relationship between $\log(\varepsilon_{emp})$ and $\log(k_{600})$ (Standard: $R^2$ = 0.16, n = 13, p = 0.19 and Flexible Foil: $R^2$ = 0.81, n = 7, p = 0.006). However, the corresponding coefficients lied outside the 95 %CI range (from 0.31 to 0.41) given by Ulseth et al. (2019). We propose that the reason that explains the observed difference between the scaling exponents derived in this paper and the values available from Ulseth et al. (2019) is represented by the specific setup of this study (i.e., an artificial

flume monitored by means of floating chambers), that differs significantly from that used by Ulseth et al. (2019) (i.e., gas tracer injections in natural streams).

### 3.6 Analysis of the uncertainty

The uncertainty in the estimate of gas-exchange velocities and equilibrium concentration via the exponential model (Method 2) was analysed using a GLUE technique (see Sect. 2.6). In most cases, the posterior bi-variate distribution was found to be

hump-shaped, peaking around the pair of values that provided the best fitting (i.e. maximum NSE, identified by the orange star in Fig. 4). However, this was not always the case. In particular, for runs that were too short to reach full saturation, we often found heterogeneous pairs of $C'_e$ and $k_{600}$ values that performed equally well. The GLUE procedure allowed us to derive a range of likely parameters associated to each saturation curve, providing a tool to characterize the robustness of the best fit obtained during the model calibration phase. It is worth noting that the GLUE procedure could be used also to analyze the

uncertainty in the estimate of $k_{600}$ obtained from the application of Method 1 to the data gathered via the drifting chambers. However, Method 1 is intrinsically associated with high uncertainties because each experiment performed in the drifting mode consisted of very few observations (see Sect.3.3). Consequently, model performances were lower than those obtained with a steady chamber, and the uncertainty was high. In particular, the standard deviation of the posterior pdf of $k_{600}$ resulting from Method 1 was typically 1 to 2 orders of magnitude higher than the posterior standard deviation obtained using Method 2.

For method 1, most of the uncertainty is induced by the lack of information on $C'_e$ during the drifting runs, which represents a direct consequence of the limited number of available $CO_2$ observations. The above result highlights at the value of long-lasting chamber runs that contain several consecutive $CO_2$ measurements in reducing the uncertainty of the estimate of $k_{600}$.

To assess the uncertainty associated to the average values of $k_{600}$ derived from Method 2 for each setup (with the Flexible Foil and Standard chambers), we calculated the average among the posterior pdfs associated to each run for a certain setup and

chamber. Practically, we considered all the experiments made with a given chamber for each setup and we took all the pairs





of parameters ($k_{600}$ and $C'_e$) that were able to reproduce (with a pre-defined minimum degree of performance) the different saturation curves available. Thus, we ended up with a bi-variate probability density density function for $C'_e$ and $k_{600}$ associated to a given threshold performance (in this case NSE$_t$ = 0.98). The analysis of the uncertainty associated to the estimate of $k_{600}$ is shown in Fig. 5, where the overall posterior pdfs of $k_{600}$ obtained across all the setups are presented for the Flexible

Foil (orange) and the Standard (blue) chamber. The mean $k_{600}$ from the best fitting procedure (Table 3) was very close to the corresponding posterior mean for all the chambers and settings. This result reinforces the robustness of the results shown in Table 3. Overall, the uncertainty in the estimate of $k_{600}$ was moderate to high (standard deviations of the posterior pdfs were in the range 1.6 to 8.2 m/d). In particular, the Standard chamber showed a mean posterior standard deviation that was 20% higher than that of the Flexible Foil chamber (see also Table 3). In setup 3 and 4 the posterior pdfs obtained with the Standard

chamber were bi-modal. Moreover, they partially overlapped with the posterior pdfs obtained with the Flexible Foil chamber in correspondence of their posterior mean. As the posterior pdf represent the likelihood of different parameter values in the light of the available observations, this overlapping indicates that the values of $k_{600}$ that allowed a good match of the data gathered with the Flexible Foil chamber (i.e. the posterior modes in Fig. 5e and 5g) provided reasonably good fit also to the saturation curves obtained through the Standard Chamber in the same configurations, at least for some runs. This result is yet another

indication that the estimates of $k_{600}$ provided by the Flexible Foil chamber were most likely more robust than the estimates obtained using a Standard Chamber.

We also analyzed the error associated to the estimate of $k_{600}$ shown in Table 3, where the mean values of $k_{600}$ resulting from all the runs of each setup are shown. The analysis consisted of calculating the maximum performance achievable for a given setup using a single value of $k_{600}$ for all the runs performed in that setup. When a single value of $k_{600}$ is calibrated against all

the runs of every setup, for some runs of the Standard Chamber the performance of the exponential model was lower than that obtained using the Flexible Foil chamber. For instance, in setup 4 we had a minimum NSE across all the runs of 0.95 for the Standard Chamber and a minimum NSE of 0.98 for the Flexible Foil Chamber. In other words, with the Standard chamber we could not find a unique set of model parameters that was able to properly fit all the individual runs performed in some setups. This is a manifestation of the limited degree of overlapping among the posterior pdfs associated to each single run of a given

setup when the Standard chamber was used.

The GLUE analysis also enabled the identification of the relative contributions to the total uncertainty in our $k_{600}$ estimate associated to: i) the uncertainty of model fitting during a single run; and ii) the uncertainty induced by heterogeneity among replicate runs. This was done by comparing the posterior pdf associated to a single run and that obtained by averaging the results of all the runs performed under the same conditions. The GLUE analysis evidenced a non-negligible uncertainty associated

to the fitting of the exponential model to the data derived from a single run. As illustrated in Fig. 5 (panels a and c), the estimate of the gas exchange velocity from a single experiment had an intrinsic uncertainty, that was about 25 % of the best fit value of $k_{600}$. Interestingly, the uncertainty in $k_{600}$ estimates associated to single runs slightly increased with increasing $\varepsilon$ in all the available cases. Furthermore, we observed that the heterogeneity among replicate experiments generated an additional contribution to the uncertainty for all the setups in which replicate experiments were performed. This uncertainty accounted

for the variability of the physical, environmental and hydrodynamic conditions experienced by the chamber during runs that





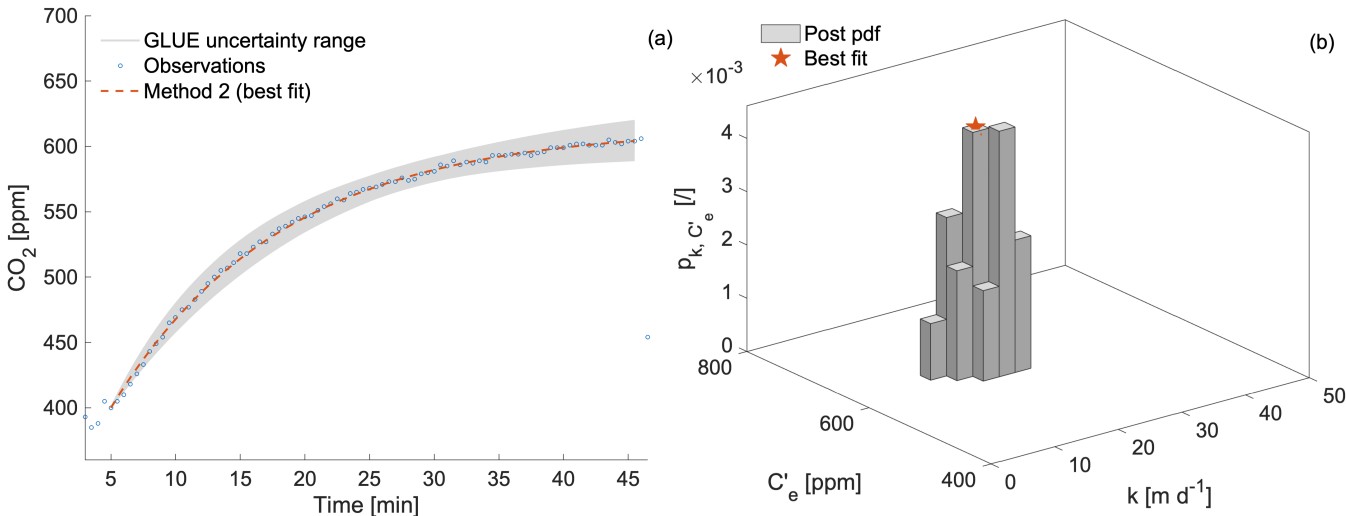

**Figure 4.** Panel (a) contains the observations of $CO_2$ with Flexible Foil chamber - steady mode - for the setup 4. The curve in orange is the result of the fitting procedure with Method 2. Panel (b) contains the bi-variate posterior pdf of $k$ and $C'_e$; the orange dot indicates the optimal parameters resulting from the fitting procedure with Method 2.

were performed at different times/dates under the same conditions (chamber's type and setup). The contribution to the total uncertainty of $k_{600}$ induced by the heterogeneity of replicate measurements markedly increased (i.e., more than linearly) with $\varepsilon$. Consequently, above a certain threshold of $\varepsilon$ the uncertainty of $k_{600}$ was dominated by the heterogeneity of the saturation curves associated to replicate measurements. This emphasizes the value of replicated experiments performed under the same

conditions in quantifying the uncertainty of $k_{600}$ estimations within systems characterized by high $\varepsilon$.

## 4   Conclusions

In this paper we analyzed the results of a flume experiment aimed at the quantification of $k_{600}$ via the application of the floating chamber methodology. During the experiment, two chamber designs (Standard vs Flexible Foil) and two deployment modes (anchored vs drifting) were compared, and the uncertainty in the estimate of the gas transfer velocity was analyzed using the

GLUE procedure. The main conclusions of the work can be summarized as follows:

    – Overall, our estimates of gas exchange velocities and water equilibrium concentrations were in line with previous results (Moog and Jirka, 1999; Zappa et al., 2007; Tokoro et al., 2008; Vachon et al., 2010; Schelker et al., 2016; Ulseth et al., 2019), with a general increase of the gas exchange velocity for larger turbulent kinetic energy dissipation rates.

    – Differently from the Standard chamber, in most setups the Flexible Foil chamber exhibited coherent results between runs

performed under the same conditions with different deployment modes (anchored or drifting). Thus we propose that the


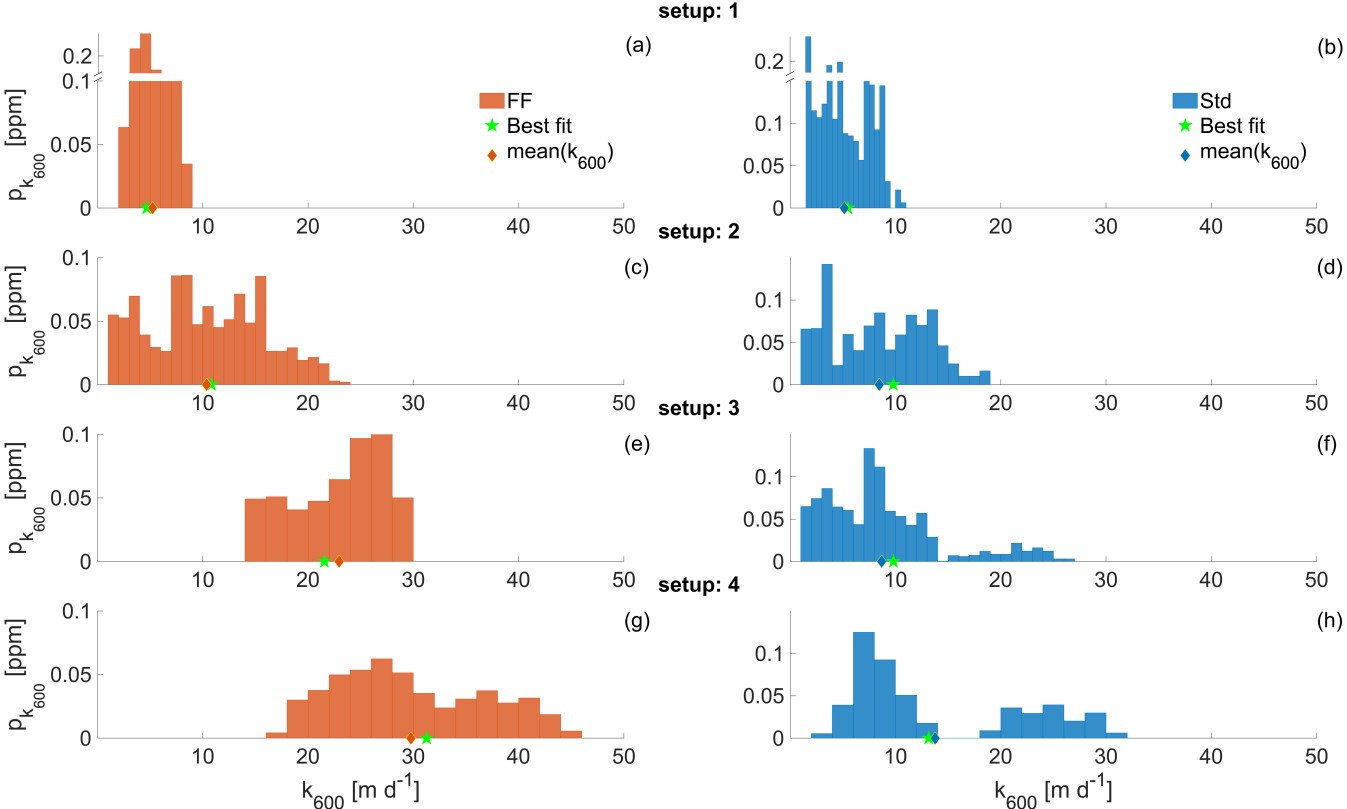

**Figure 5.** Overall posterior marginal pdf of $k_{600}$ for each different setup (1, 2, 3, 4) and for both Flexible Foil (orange: panels a, c, e, g) and Standard (blue: panels b, d, f, h) chambers. The pdfs were obtained using a threshold performance (NSE$_t$) of 0.98. The green stars indicate the best fit from Table 3, while the diamonds represent the posterior averages.

Flexible Foil chamber can ensure a greater flexibility of use (drifting or anchored deployments) across a wide range of field conditions.

– The Flexible Foil chamber gave consistent $k_{600}$ responses to changes in the slope and/or discharge in all the setups. Moreover, the ADV measurements carried out below the chambers during the runs indicated that the estimate of $k_{600}$ was biased by only $+6\%$ owing to the impact of the Flexible Foil chamber on the underlying flow field. Conversely, the bias increased to $+18\%$ when the Standard chamber was used. The limited turbulence induced by our Flexible Foil chamber and its ability to replicate previously observed relationships between $k_{600}$ and slope/discharge/kinetic energy makes it a promising tool for future stream $CO_2$ studies.

– In our experiment, uncertainty in the estimate of gas exchange velocity was moderate to high, with a standard deviations of $k_{600}$ between 1.6 and 8.2 m/d for the anchored mode. Drifting deployments, instead, were typically characterized by a larger uncertainty, with a huge increase in the uncertainty of $k_{600}$ estimates mainly due to the lack of information







on $C'_e$ during the drifting runs. While drifting deployments allow useful estimates of spatially-integrated $k_{600}$ within whole river segments, our results highlighted the value of steady runs for limiting the uncertainty of water equilibrium concentrations and gas exchange velocities in streams.

– The comparison of the uncertainty in $k_{600}$ estimates obtained using different chamber types evidenced that a smaller uncertainty is associated to the Flexible foil chamber ($-$ 20% with respect to the Standard Chamber). Likewise, when a single $k_{600}$ is associated to all the runs of a given setup, the exponential model had a poorer fitting to the data obtained with the Standard Chamber as compared to the Flexible Foil chamber, because of the enhanced heterogeneity of the different runs performed with the Standard Chamber.

– Our study highlighted the importance of quantifying different sources of errors in the estimate of gas exchange velocities derived via the chamber methodology. While the uncertainty in $k_{600}$ estimates was dependent on the chamber design and the deployment technique, in general uncertainty was higher in systems with high turbulent kinetic energy dissipation rate, where the heterogeneity among replicate experiments dominates. Thus, we propose that performing different replicate experiments under the same conditions should become a standard practice in the quantification of the magnitude 440 (and the uncertainty) of gas exchange velocities, particularly in high-energy hydrologic systems.

In conclusion, this study indicated that the Flexible Foil chamber used in an anchored deployment mode is able to enhance the reliability and decrease the uncertainty of $CO_2$ measurements in low-order streams. Furthermore, the analyses shown in the paper allowed us to identify an objective procedure to properly handle model parameter uncertainty in $CO_2$ outgassing studies based on the chamber methodology. These findings can be easily generalized to other soluble gases transported by non-bubbly 445 flows, with potentially important implications for the estimate of gas exchange processes in running water.

## Appendix A: Turbulent kinetic energy dissipation estimation from ADV measurements

In order to compute $\varepsilon$, flow velocities were measured in four directions using an Acoustic Doppler Velocity (ADV) meter (Vectrino+, downlooking probe, Nortek, Rud, Norway). For each experiment, we took 24 measurements along 6 transects 450 along the flume profile, with 4 measurements per transect. The probe was sampled at 200 Hz, with a preset velocity range of 0.1 to 0.3 m or 0.3 to 1 m, a sampling volume height of 7 mm and a transmit length of 1.8 mm. The probe was mounted to a custom-made frame and tilted (heading=0°, roll=45°, pitch=57°) to allow measurements at 3.5 cm below the water surface. We rotated the velocities to an earth coordinate system with an along-flow ($u$), cross-flow ($v$) and two duplicate up-down ($w_1$, $w_2$) components following standard transformations provided by Nortek (2020). To increase the scatter of the acoustic 455 pulse emitted from the probe, we enhanced the turbidity of the inflowing water by adding a solution of wheat flour and water (120 $\mathrm{g\,l^{-1}}$) at constant flow rates (50-80 $\mathrm{ml\,min^{-1}}$ using a peristaltic pump (Schlauchpumpe MV-GE, Ismatec Cole-Parmer GmbH, Wertheim, Germany). This setup allowed nearly spike-free recordings with signal-to-noise ratios (SNR) of 21.5±2.1 (mean±Std) and correlations of 92.2±5.2%. We rejected time series with SNR < 15 and correlations < 70. The few spikes





remaining in accepted time series were removed following Goring and Nikora (2002). Spikes were defined as measurements
that differed by more than 4 times the standard deviation of the difference between the raw time series and its running median
(window length 1 s). We replaced those measurements by the running median. We estimated $\varepsilon$ using the inertial dissipation
method following Zappa et al. (2003) with modifications by Bluteau et al. (2011). Accordingly, the kinetic energy of isotropic
turbulence is dissipated by breaking larger to smaller eddies within the intertial subrange of wavenumbers. $\varepsilon$ can be derived
from the wavenumber spectrum (S) of the fluctuating velocities of component i $\in \{u, v, w_1, w_2\}$ as:

$$S_i(\kappa) = \alpha_i c \varepsilon_i^{2/3} \kappa^{-5/3} \tag{A1}$$

where $c = 1.5$ is the empirical Kolmogorov universal constant, $\alpha_{(i=1)} = \frac{18}{55}$ for the component in the direction of mean advec-
tion, $\alpha_{(i>1)} = 1.33\frac{18}{55}$ in the other directions perpendicular ($i = 2$) or vertical ($i = 3$) to the direction of mean advection (Pope,
2000), $\kappa = \frac{2\pi f}{u_{adv}}$ is the wave number, $f$ is the frequency and $u_{adv}$ is the mean advection velocity. We derived the wavenumber
spectra of fluctuating velocities from the frequency spectra, assuming Taylor's hypothesis of frozen turbulence, i.e. turbulent
motions, quantified by the root mean square of fluctuating velocities ($s_i$), are small relative to $u_{adv}$. We calculated $u_{adv}$ by
rotating the velocities such that $v = w_1 = w_2 = 0$ following Foken (2008). In our experiment $\frac{s_i}{u_{adv}^3}$ was always $< 0.04$, giving
strong support to the Frozen turbulence hypothesis (Kitaigorodskii and Lumley, 1983). We generated the frequency spectra
by means of the Welch's method using 8 segments with 50% overlap and tapered with a Hamming window. We corrected the
measured frequency spectra for pulse averaging by dividing them by $[a_1(f) + a_2(f)]$, where:

$$a_1(f) = \left(\frac{sin(\pi f \Delta t)}{\pi f \Delta t}\right)^2 \tag{A2}$$

$$a_2(f) = \left(\frac{f}{f_0 - f}\right)^{5/3} \left(\frac{sin(\pi(f_0 - f)\Delta t)}{\pi(f_0 - f)\Delta t}\right)^2 \tag{A3}$$

and $f_0 = 200$ Hz is the nominal sampling frequency (Henjes et al., 1999). To find the intertial subrange within which $\varepsilon$ is
calculated, we used an approach following recommendations by Bluteau et al. (2011). Specifically, we fitted Eq. (5) to the
measured spectra within wave number intervals of different widths and locations along the wave number axis. To define these
'candidate' intervals, we evaluated all possible combinations of lower and upper interval bounds. We set the minimum lower
bound set to $2\pi/D$ and the maximum upper bound to $2\pi/L$, where $D$ is the local water depth (m) and $L$ is the length scale of
the ADV sampling volume (m$^3$) (c.f. Zappa et al., 2003). We also required $S$ to drop by at least one order of magnitude within
the intervals (Bluteau et al., 2011). For each 'candidate' interval, we modelled the wave number spectrum using maximum
likelihood estimation following Bluteau et al. (2011), assuming that the ratio of observed ($\widetilde{S}$) and modelled ($S$) spectral esti-
mates follows a $\chi_{df}^2$ distribution ($df\frac{\widetilde{S}}{S} = \chi_{df}^2$) where $df$ represents the degrees of freedom of the system. Using the probability
density function of the $\chi_{df}^2$ distribution:

$$f(a) = \frac{a^{(d-2)/2} \exp(-a/2)}{2^{d/2}\left(\frac{d-2}{2}\right)!}, a \geq 0 \tag{A4}$$





we computed the log-likelihood for spectral observations

$$\ln L = n \ln d - \sum_{j=1}^{n} \ln S_{ij} + \sum_{j=1}^{n} \ln f(a_j), \tag{A5}$$

so as to find the most likely estimate for $\epsilon$ for the specific wave number interval. We determined the minimum 95% confidence interval of $\varepsilon$ as $\pm 1.96 \sqrt{\mathrm{var}(\varepsilon)}$ where $\mathrm{var}(\varepsilon)$ is the variance of $\varepsilon$ calculated based on the curvature of the maximum log likelihood

$$\mathrm{var}(\varepsilon) \geq \frac{-1}{\frac{\delta^2 (\ln L)}{\delta \varepsilon^2}} \tag{A6}$$

To evaluate the goodness of fit for each 'candidate' interval, we computed the maximum absolute deviation (MAD)

$$MAD = \left| \frac{1}{n} \sum_{j=0}^{n} \left( \frac{\widetilde{s}_{ij}}{s_{ij}} - \left\langle \frac{\widetilde{s}_i}{s_i} \right\rangle \right) \right| \tag{A7}$$

following Ruddick et al. (2000). The interval that yielded the lowest MAD was used to determine the final $\varepsilon$ estimate. As an additional measure of the goodness-of-fit, we calculated the coefficient of determination:

$$R^2 = 1 - \frac{\sum_{j=0}^{n} \left( \widetilde{s}_{ij} - s_{ij} \right)^2}{\sum_{j=0}^{n} \left( \widetilde{s}_{ij} - \langle s_{ij} \rangle \right)^2} \tag{A8}$$

We rejected all $\varepsilon$ estimates with $MAD > 2(\frac{2}{d})^{1/2}$ or $R^2 < 0$. For comparison with flume-integrated $k_{600}$ values, we calculated the arithmetic mean of all $\varepsilon$ estimates of the vertical component $\varepsilon_{w_1}$ if the minimum 95% confidence intervals of these estimates and the corresponding estimates for the other directions ($\varepsilon_u$, $\varepsilon_v$) overlapped, thereby implying that the assumption of isotropic turbulence was fulfilled.

Code to calculate epsilon from the ADV data is provided under the GPL-3.0 License at https://github.com/MarcusKlaus/CalculateEpsilon.

## Appendix B: Standardisation of the gas exchange velocity

In order to compare the results of $k$ with other gas exchange velocities at 20 °C, the observed values of $k$ were converted to a standardised gas exchange rate $k_{600}$ as:

$$k_{600} = k \left( \frac{600}{Sc_{CO_2}} \right)^{-n} \tag{B1}$$

where the subscript 600 indicates the Schmidt number of $CO_2$ at 20 °C in freshwater. The exponent $n$ in Eq. (B1) is usually between 1/2 and 1; 1/2 is more appropriate for rough surfaces, 2/3 for calm waters (Rawitch et al., 2019). Therefore, 1/2 is a proper value for our analyses. The Schmidt number ($Sc$) is a dimensionless number expressing the ratio between the kinematic viscosity of water to the diffusivity of gas. $Sc$ of $CO_2$ ($Sc_{CO_2}$) can be estimated from water temperature ($T_\mathrm{w}$) via the following expression (Raymond et al., 2012):

$$Sc_{CO_2} = 1742 - 91.24 T_w + 2.208 T_w^2 - 0.0219 T_w^3 \tag{B2}$$

Standardisation to $k_{600}$ requires the knowledge of the water temperature which was unknown during the experiment. In order to overcome this issue, we decided to use a unique time/temperature relationship derived from the interpolation of the few available data even though we are conscious of the possible biases because of the change in daily temperature cycles during the experiment. Given the observed maximum diurnal water temperature excursion (10-15 °C), the possible influence of errors 520 in temperature on the $k_{600}$ estimate is limited ($\pm$ 8%).

### Appendix C: Design of the Flexible Foil chamber

Figure C1 contains three photos from different perspectives of the Flexible Foil chamber. An overview of Flexible Foil chamber (Fig. C1a) shows the floating system designed via three half-liter water bottles. In order to have a real time control on the measurement a top casing containing the USB connection to the sensor was developed (Fig. C1a). The rigid cup of the chamber 525 was sustained above the water surface by the floating system that was fixed on the external cup margin (Fig. C1b). The flexible sealing via polyethylene foil (Plastic sheet UV4, Foliarex, Poznań, Poland) and adhesive tape (Extra Power Universal, Tesa, Hamburg, Germany) was extended all around the lower internal profile of the chamber cup (Fig. C1b). The sealing guaranteed a complete isolation of the air inside the chamber from the external atmosphere. An open protection of the $CO_2$ sensor was developed to prevent contact with water droplets (Fig. C1c).

*Author contributions.* All authors contributed to design and perform the experiment in Lunz. ND designed and built the flexible foil chamber. FV analyzed the chamber results, and wrote a first draft of the paper. GB identified the methods for the analysis and, together with MK and JS, provided insight into data interpretation. MK performed the ADV measurements and analyzed the ADV data. All authors contributed to finalizing and editing the paper.

*Competing interests.* The authors declare that they have no conflict of interest.

*Acknowledgements.* This study was supported by the European Research Council (ERC) DyNET project funded through the European Community's Horizon 2020 - Excellent Science - Programme (grant agreement H2020-EU.1.1.-770999). This study was also supported by the Transnational Access grant ExSONIC though the European Commission EU H2020-INFRAIA-project AQUACOSM (Grant No 731065). We thank Michael Scharner and Paul Schweigerlehner for field assistance, Gertraud Stenizcka for logistical assistance, and Johann Waringer for providing the ADV system.





**Figure C1.** Overview of the Flexible Foil chamber (panel a). Close up on the support of the floating system (panel b). Protection of the sensor from the water droplets (panel c).

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
