# Peer review of "Evaluating stream CO2 outgassing via Drifting and Anchored flux chambers in a controlled flume experiment"

_Biogeosciences, 2020_

## Referee Comment (RC1) · Anonymous Referee #1 · 26 Oct 2020

The authors assess different types of flux chambers and different modes of deployment for measuring CO2 fluxes and corresponding gas exchange velocities in running waters. They performed a set of measurements in laboratory flumes, where flow velocity and slope could be adjusted, and compared static chambers (held at a fixed locations), drifting chambers (drifting with the flow), regular chambers (where the chamber edges penetrate into the water) and flexible foil chambers (chamber is sealed to the atmosphere using an adhesive foil). The main focus of their analysis is on estimating gas exchange velocities (k600), which are estimated from measured fluxes and dissolved gas concentrations. Exchange velocities are primarily controlled by near-surface turbulence in water, which has been measured during the experiments.

[Figure]

In line with previous research, the study results suggest that application of standard chambers in results in chamber artifacts, particularly in static deployments. The novel aspect of the present study is in the detailed analysis of systematic and statistical errors when chamber measurements are used for estimating gas exchange velocities. Unfortunately, there have been no reference measurements of fluxes and exchange velocities in the present study, so that the conclusions are based on comparisons of chamber performance. Nevertheless, the manuscript fills an important gasp, as the effect of chamber design and deployment mode on measuring gas fluxes from aquatic systems received very little scientific attention.

The manuscript is very well written and easy to follow. I have a few suggestions and technical comments, which may improve clarity:

- dissipation rates of turbulent kinetic energy were estimated using a bulk approach (from channel slope and flow depth), as well as from local measurements of turbulent velocity fluctuations. I suggest to add and to discuss a comparison of both dissipation rate estimates, as the bulk approach can be more easily applied to field conditions.

- I suggest to mention the range and variability of measured $CO_2$ fluxes 8in addition to dissolved concentration and gas exchange velocity.

- Fig. 2: why not using a scaled x-axis (instead of a categorical axis), where significant regressions could be added to the graph.

- Comparison of the scaling coefficient (alpha) in the equation relating k600 to dissipation rates to other studies: energy dissipation rates depend on the depth at which measurements were taken (see e.g. Esters et al. 2017, doi: 10.1002/2016JC012088 for wind-driven systems). In streams, turbulence is driven by bed friction, leading to a different depth-dependence of dissipation rates (and values of the "constant" alpha). Dissipation rates from bulk scaling (see above), in contrast, assume uniform distributions. This issue could be discussed further when comparing the results to other studies.

- line 452: I assume that the ADV velocities were rotated into a vertically-oriented coordinate system before all subsequent analyses? line 483: lower bound of the wave number for spectral fitting: the lower bound should not be defined by water depth, but by the distance of the ADV sampling volume from the water surface (as this defined the larges isotropic eddy).

- line 501: estimates of energy dissipation rates are typically log-normally distributed (see e.g., Baker et al. 1987, https://doi.org/10.1175/1520-0485(1987)017<1817:STITSO>2.0.CO;2). Arithmetic averaging may therefore be not appropriate.

---

## Referee Comment (RC2) · Anonymous Referee #2 · 27 Oct 2020

The study by Vingiani et al. compares gas transfer velocities (k600) using 2 different chamber designs (Flexible Foil and Standard) under different conditions (drifting vs anchored) to evaluate accuracy and uncertainty of k600 in rivers using the chamber method. The authors conclude that the flexible foil chamber may be a useful tool to estimate k600 under anchored conditions in low-order streams. The study fills an important research gap that will be of great interest to the scientific community. The manuscript is well organised and well written. My comments are mainly in regards to introduction and methods and intended to help to improve this manuscript. The results and discussion section were thought through and easy to follow.

[Figure]

L26 greenhouse gas emissions

L33 might be helpful to show Eq. 1 here

L37 Yes, k can vary in space and time, which is a very important characteristic. I suggest to expand the aspect of spatiotemporal heterogeneity. Maybe the authors can add some examples or numbers to give us a better understanding how much k can vary in space and time in rivers? This could be then also used in the discussion of spatial k600 of drifting k600.

L40 Are there k models for rivers other than from Raymond et al. 2013? If yes, do they also use wind, current and slope? I'm surprised to see only one reference here.

L55 a floating "flying" chamber design with flexible chamber walls has also been successfully used by Rosentreter et al. 2017 and Jeffrey et al. 2018

L58 Yes, local CO2 sources such as groundwater inputs change surface water CO2 concentration, but how would they interfere with local k?

L69-70 This may be exaggerated. For example, the study by Rosentreter et al. 2018 compared k of CO2 in mangrove surrounded creeks, lakes, main river channel, and a bay and in direct comparison to a dual tracer experiment and found good agreement between the two methods (5% discrepancy). Lorke 2015 compared drifting vs anchored chamber measurements. Jeffrey et al. 2018 compared chamber measurements in different sections of an estuary. etc... so this has been discussed before and also quantified.

Was there a fan attached inside the two chambers? Did you test for evenly distributed air circulation inside the chamber?

Did you test for temperature artefacts inside the chamber? Was the temperature constant during chamber incubations?

L126 what CO2 sensors? Please add brand, model, and accuracy of CO2 sensor and

CO2 analysis.

L127 Roughly, how long did you conduct chamber incubations (runs) for? minutes, half an hour? an hour?

L134-135 Is this a problem? Even if chamber concentrations inside were not atmospheric, you can still use the change of concentration for estimating k, no? If you measured CO2 every 30sec over the duration of the chamber incubation, then you have a start and end concentration over time (F) that you applied in Eq.2 and Eq.3. Meaning only the difference between start and end concentration is important (slope) and not the concentration itself. I'm curious to hear if the authors agree or disagree.

L139 Were the atmospheric concentrations outside close to 400 ppm?

L193 do you mean increasing "linear regression"? If yes, what was your threshold r2?

L250-251 this sentence could be deleted as this is also mentioned in the Table 3 caption.

Figure 4b shouldn't this be k600, not k?

While this study greatly contributes to our understanding of appropriate chamber design and conditions (drifted vs anchored) of the chamber method in general, I wonder how good this chamber method is in predicting the CO2 flux in comparison to other k methods and empirical k models? For example, were CO2 fluxes measured in the flumes better predicted by k600 derived from the chambers measurements in this study than predicted from k600 models (e.g. Raymond et al. 2013, Ulseth et al. 2020)? Or more practically, would the authors recommend to use FF chamber anchored mode over the k600 model by Ulseth et al. (2020) based on energy dissipation for estimating CO2 fluxes in rivers? Do the empirical models under or overestimate the flux?

---

## Author Comment (AC1) · 16 Nov 2020

We thank the anonymous Reviewer for her/his considerations/comments. We will be happy to improve the manuscript based on the suggestions and comments provided by the referee. A more detailed answer to each specific comment follows.

Comment: Dissipation rates of turbulent kinetic energy were estimated using a bulk approach (from channel slope and flow depth), as well as from local measurements of turbulent velocity fluctuations. I suggest adding and to discuss a comparison of both dissipation rate estimates, as the bulk approach can be more easily applied to field conditions. Answer: Thank you. We will add a brief discussion about this issue.

Comment: I suggest mentioning the range and variability of measured CO2 fluxes in addition to dissolved concentration and gas exchange velocity. Answer: We will insert the range of CO2 fluxes in the text.

Comment: Fig. 2: why not using a scaled x-axis (instead of a categorical axis), where significant regressions could be added to the graph. Answer: We initially used a scaled x-axis for velocity and discharge but then we decided it was better to use a categorical one to have a clear visual comparison of the mean values obtained with the different designs and deployments. Linear regressions for slope and discharge were computed and discussed in the text but not shown in the plot (see L266-282).

Comment: Comparison of the scaling coefficient (alpha) in the equation relating k600 to dissipation rates to other studies: energy dissipation rates depend on the depth at which measurements were taken (see e.g. Esters et al. 2017, doi: 10.1002/2016JC012088 for wind-driven systems). In streams, turbulence is driven by bed friction, leading to a different depth-dependence of dissipation rates (and values of the "constant" alpha). Dissipation rates from bulk scaling (see above), in contrast, assume uniform distributions. This issue could be discussed further when comparing the results to other studies. Answer: We agree, we will add a more complete discussion about this issue in the revised text.

Comment: line 452: I assume that the ADV velocities were rotated into a vertically oriented coordinate system before all subsequent analyses? Answer: Yes, we agree. We will add a sentence to explain the rotation, e.g. "We rotated the velocities to an earth coordinate system with an along-flow (u), cross-flow (v) and two duplicate up-down (w1, w2) components following standard transformations provided by Nortek (2020)."

Comment: line 483: lower bound of the wave number for spectral fitting: the lower bound should not be defined by water depth, but by the distance of the ADV sampling volume from the water surface (as this defined the larges isotropic eddy). Answer: Thank you very much for pointing this out. We agree on this point and we will correct

the Ms accordingly.

Comment: line 501: estimates of energy dissipation rates are typically log-normally distributed (see e.g., Baker et al. 1987, https://doi.org/10.1175/1520-0485(1987)017<1817:STITSO>2.0.CO;2). Arithmetic averaging may therefore be not appropriate. Answer: Thank you very much for pointing this out. We have verified that our data was log-normally distributed.
* * *

---

## Author Comment (AC2) · 16 Nov 2020

We thank the anonymous Reviewer for her/his considerations/comments. We will be happy to improve the manuscript based on the suggestions and comments provided by the referee. A more detailed answer to each specific comment follows.

Comment: L26 greenhouse gas emissions Answer: Thank you. We will correct it.

Comment: L33 might be helpful to show Eq. 1 here Answer: Ok, we will fix it.

Comment: L37 Yes, k can vary in space and time, which is a very important characteristic. I suggest expanding the aspect of spatiotemporal heterogeneity. Maybe the

authors can add some examples or numbers to give us a better understanding how much k can vary in space and time in rivers? This could be then also used in the discussion of spatial k600 of drifting k600. Answer: Thank you for this suggestion. We will expand the issue of spatiotemporal heterogeneity of K in the revised text.

Comment: L40 Are there k models for rivers other than from Raymond et al. 2013? If yes, do they also use wind, current and slope? I'm surprised to see only one reference here. Answer: In L40 we quoted the paper from Raymond et al. 2012 (not Raymond et al. 2013). In this paper the authors summarized many of the available equations relating gas transfer velocity to the hydraulic geometry. We will replace "Raymond et al. 2012" with "Raymond et al. 2012 and references therein"

Comment: L55 a floating "flying" chamber design with flexible chamber walls has also been successfully used by Rosentreter et al. 2017 and Jeffrey et al. 2018 Answer: We will add the suggested references. Thank you.

Comment: L58 Yes, local CO2 sources such as groundwater inputs change surface water CO2 concentration, but how would they interfere with local k? Answer: Local groundwater inputs rich in CO2 do not affect k. We argued that the chamber method allows direct point measurements of gas fluxes and these latter could be helpful to observe the spatial heterogeneity of gas fluxes in a stream sourced by pointy CO2 groundwater inputs. We will better clarify this point in the revised text.

Comment: L69-70 This may be exaggerated. For example, the study by Rosentreter et al. 2018 compared k of CO2 in mangrove surrounded creeks, lakes, main river channel, and a bay and in direct comparison to a dual tracer experiment and found good agreement between the two methods (5% discrepancy). Lorke 2015 compared drifting vs anchored chamber measurements. Jeffrey et al. 2018 compared chamber measurements in different sections of an estuary. etc... so this has been discussed before and also quantified. Was there a fan attached inside the two chambers? Did you test for evenly distributed air circulation inside the chamber? Did you test for temperature

artefacts inside the chamber? Was the temperature constant during chamber incubations? Answer: All the studies mentioned in this comment (Rosentreter et al. 2018, Lorke et al. 2015, Jeffrey et al. 2018) discuss or quantify the uncertainty associated to the estimate of k via the chamber method but using the water $CO_2$ concentrations obtained via other techniques. In our manuscript we discuss and quantify the uncertainty associated to the estimates of k derived only from $CO_2$ observations inside the chamber. This will be further stressed in the revision of the paper. A fan was not used inside the chamber and the temperature was always almost constant during the chamber incubation. This information will be added in the Method section.

Comment: L126 what $CO_2$ sensors? Please add brand, model, and accuracy of $CO_2$ sensor and $CO_2$ analysis. Answer: The $CO_2$ sensor specifics are indicated in L101:104.

Comment: L127 Roughly, how long did you conduct chamber incubations (runs) for? minutes, half an hour? an hour? Answer: Chamber incubation time for steady deployments ranges between 12 mins to 45 mins. This will be specified in the revised methods.

Comment: L134-135 Is this a problem? Even if chamber concentrations inside were not atmospheric, you can still use the change of concentration for estimating k, no? If you measured $CO_2$ every 30sec over the duration of the chamber incubation, then you have a start and end concentration over time (F) that you applied in Eq.2 and Eq.3. Meaning only the difference between start and end concentration is important (slope) and not the concentration itself. I'm curious to hear if the authors agree or disagree. Answer: The equilibration condition at the moment of incubation does not affect per se the measure of k. Thus, we agree on the possibility of estimating k also in the case the chamber is not perfectly equilibrated to the atmospheric value at the moment of incubation. In general, ensuing a perfect equilibration before incubation is important to have a realistic representation of the $CO_2$ fluxes occurring at the water-atmosphere interface. Moreover, if the chamber is not equilibrated to the atmosphere you might

have a non-homogeneous $CO_2$ concentration inside the chamber at the beginning of the incubation and this could generate a potential bias in the typical exponential curve during the incubation (i.e. Eq. 2 and 3 could be no longer valid). Also, ensuing equilibration to the atmospheric value before incubation guarantees the maximization of the concentration gradient and this leads to a clearer $CO_2$ signal.

Comment: L139 Were the atmospheric concentrations outside close to 400 ppm? Answer: Yes, we will explicitly indicate it around L139.

Comment: L193 do you mean increasing "linear regression"? If yes, what was your threshold r2? Answer: The first data quality check we used was to consider only the curves that showed a monotonous increase (or decrease) in $CO_2$ concentration inside the chamber volume during the incubation process. This was just a visual data quality check. Then we used the NSE coefficient to discriminate the ones with the highest performance. This wil be clarified in the revised Methods.

Comment: L250-251 this sentence could be deleted as this is also mentioned in the Table 3 caption. Answer: We will delete it. Thank you.

Comment: Figure 4b shouldn't this be k600, not k? Answer: No, the figure is referred to a single deployment (there is no need to standardize the k to k600 in this case).

Comment: While this study greatly contributes to our understanding of appropriate chamber design and conditions (drifted vs anchored) of the chamber method in general, I wonder how good this chamber method is in predicting the $CO_2$ flux in comparison to other k methods and empirical k models? For example, were $CO_2$ fluxes measured in the flumes better predicted by k600 derived from the chambers measurements in this study than predicted from k600 models (e.g. Raymond et al. 2013, Ulseth et al. 2020)? Or more practically, would the authors recommend to use FF chamber anchored mode over the k600 model by Ulseth et al. (2020) based on energy dissipation for estimating $CO_2$ fluxes in rivers? Do the empirical models under or overestimate the flux? Answer: We thank you for your comment. In this paper, we do not have a direct

comparison between the chamber method and other possible k methods. Hence, we are not able to state that the chamber method might be better than other k methods. In general, we observed k estimates from FF chamber in line with empirical models from Raymond et al. 2013 and Ulseth et al. 2020. We are not able to argue that our chamber estimates under or overestimates the flux with respect to these models. Your questions are all valuable, and they provide exciting hints for further research. We are conscious that to have measure of k from other methods might improve the paper and give a further support to the validity of our chamber method. We hope to have the possibility to investigate this further.
* * *

---

## Author Response (AR1)

We thank the anonymous Reviewers for their considerations/comments. We hope to have improved the manuscript based on the suggestions and comments provided by the referees.

General comment: We revised our methodology to calculate epsilon in response to the reviewer comments. We also carried out additional minor changes to the methodology as outlined below. The revisions lead to only slight changes in epsilon estimates with no major consequence for the results or conclusions of this manuscript.

A more detailed answer/change to each specific comment follows.
Note that in the "change" section we referred to the lines in the manuscript with tracked changes.

**Reviewer 1**
Comment:
Dissipation rates of turbulent kinetic energy were estimated using a bulk approach (from channel slope and flow depth), as well as from local measurements of turbulent velocity fluctuations. I suggest adding and to discuss a comparison of both dissipation rate estimates, as the bulk approach can be more easily applied to field conditions.
Answer:
Thank you. We will add a brief discussion about this issue.
Change:
We have expanded the 3.5 Section with a discussion about this issue (see L. 369-374).

Comment:
I suggest mentioning the range and variability of measured CO2 fluxes in addition to dissolved concentration and gas exchange velocity.
Answer:
We will insert the range of CO2 fluxes in the text.
Change:
We added the range of measured fluxes and associated variability in the first paragraph of the result and discussion section (see L. 278-282).

Comment:
Fig. 2: why not using a scaled x-axis (instead of a categorical axis), where significant regressions could be added to the graph.
Answer:
We initially used a scaled x-axis for velocity and discharge but then we decided it was better to use a categorical one to have a clear visual comparison of the mean values obtained with the different designs and deployments. Linear regressions for slope and discharge were computed and discussed in the text but not shown in the plot (see L. 266-282).
Change:
We have not made any change here.

Comment:
Comparison of the scaling coefficient (alpha) in the equation relating k600 to dissipation rates to other studies: energy dissipation rates depend on the depth at which measurements were taken (see e.g. Esters et al. 2017, doi:

10.1002/2016JC012088 for wind-driven systems). In streams, turbulence is driven by bed friction, leading to a different depth-dependence of dissipation rates (and values of the "constant" alpha). Dissipation rates from bulk scaling (see above), in contrast, assume uniform distributions. This issue could be discussed further when comparing the results to other studies.

Answer:

We agree, we will add a more complete discussion about this issue in the revised text.

Change:

We have added the sentences: " According to Esters et al. (2017), the coefficient α follows the strong depth-dependency in ε. While Esters et al. (2017) refer to standing waters, this can also be expected to be true in running waters were epsilon varies with depth as a result of bottom friction (Sukhodolov et al., 1998). Therefore, an in-depth comparison between our results and previous studies proves difficult because of the different measuring depths across the studies. (see L. 356-360)."

For instance, Zappa et al. (2007) measured ε from few cm to 3 m below the surface, instead Vachon et al. (2010) used ε at a depth of 10 cm.

Comment:

line 452: I assume that the ADV velocities were rotated into a vertically oriented coordinate system before all subsequent analyses?

Answer:

Yes, we agree. We will add a sentence to explain the rotation, e.g. "We rotated the velocities to an earth coordinate system with an along-flow (u), cross-flow (v) and two duplicate up-down (w1, w2) components following standard transformations provided by Nortek (2020)."

Change:

To clarify the rotation further, we added "vertically-oriented" to "earth coordinate system" (see L. 493).

Comment:

line 483: lower bound of the wave number for spectral fitting: the lower bound should not be defined by water depth, but by the distance of the ADV sampling volume from the water surface (as this defined the larges isotropic eddy).

Answer:

Thank you very much for pointing this out. We agree on this point and we will correct the Ms accordingly.

Change:

We recalculated epsilon based on the distance of the ADV sampling volume from the water surface used as the lower bound of the wavenumber for spectral fitting. We changed the methods description (L. 518) and online R code accordingly.

Comment:

line 501: estimates of energy dissipation rates are typically log-normally distributed (see e.g., Baker et al. 1987, https://doi.org/10.1175/1520-0485(1987)017<1817:STITSO>2.0.CO;2). Arithmetic averaging may therefore be not appropriate.

Answer:

Thank you very much for pointing this out. We have verified that our data was log-normally distributed.

Change:

We now use the geometric mean to characterize the central tendency in epsilon values across the flume, following Baker and Gibson (1987). We clarify this in the methods description (L. 537-540). We also rephrased the text slightly to further improve clarity.

**Reviewer 2**
Comment:
L26 greenhouse gas emissions
Answer:
Thank you. We will correct it.
Changes:
We have changed Green House Gas with greenhouse gas (see L. 26).

Comment:
L33 might be helpful to show Eq. 1 here
Answer:
Ok, we will fix it.
Changes:
We have moved the Equation 1 and its description in the introduction section. We have slightly adapt the test to fit this rearrangement (see L. 33-37 and L. 155-159).

Comment:
L37 Yes, k can vary in space and time, which is a very important characteristic. I suggest expanding the aspect of spatiotemporal heterogeneity. Maybe the authors can add some examples or numbers to give us a better understanding how much k can vary in space and time in rivers? This could be then also used in the discussion of spatial k600 of drifting k600.
Answer:
Thank you for this suggestion. We will expand the issue of spatiotemporal heterogeneity of K in the revised text.
Change:
We have added the following two sentences that provide two examples of the variability of k600 in space and time (see L. 43-46).
"For instance, Jeffrey et al. (2018) observed a 230-fold variation in the gas exchange velocity between two nearby stations along an estuary in the mid-coast of New South Wales, Australia. Likewise, Natchimuthu et al. (2017) estimated that flood events could produce a 7-fold increase of k in a hemiboreal catchment in Southwest Sweden."

Comment:
L40 Are there k models for rivers other than from Raymond et al. 2013? If yes, do they also use wind, current and slope? I'm surprised to see only one reference here.
Answer:
In L40 we quoted the paper from Raymond et al. 2012 (not Raymond et al. 2013). In this paper the authors summarized many of the available equations relating gas transfer velocity to the hydraulic geometry. We will replace "Raymond et al. 2012" with "Raymond et al. 2012 and references therein"
Change:
We have changed the references (see L. 47).

Comment:

L55 a floating "flying" chamber design with flexible chamber walls has also been successfully used by Rosentreter et al. 2017 and Jeffrey et al. 2018.

Answer:
We will add the suggested references. Thank you.

Change:
We have inserted the suggested references (see L. 64).

Comment:
L58 Yes, local CO2 sources such as groundwater inputs change surface water CO2 concentration, but how would they interfere with local k?

Answer:
Local groundwater inputs rich in CO2 do not affect k. We argued that the chamber method allows direct point measurements of gas fluxes and these latter could be helpful to observe the spatial heterogeneity of gas fluxes in a stream sourced by pointy CO2 groundwater inputs. We will better clarify this point in the revised text.

Change:
We have rephrased the sentence (see L. 68-69) with "Furthermore, chambers allow direct point measurements of gas fluxes, that are especially useful in headwater streams typically characterised by spatially heterogeneous conditions and complex CO2 patterns (Ploum et al., 2018; Rawitch et al., 2019)."

Comment:
L69-70 This may be exaggerated. For example, the study by Rosentreter et al. 2018 compared k of CO2 in mangrove surrounded creeks, lakes, main river channel, and a bay and in direct comparison to a dual tracer experiment and found good agreement between the two methods (5% discrepancy). Lorke 2015 compared drifting vs anchored chamber measurements. Jeffrey et al. 2018 compared chamber measurements in different sections of an estuary. etc... so this has been discussed before and also quantified.

Was there a fan attached inside the two chambers? Did you test for evenly distributed air circulation inside the chamber?

Did you test for temperature artefacts inside the chamber? Was the temperature constant during chamber incubations?

Answer:
All the studies mentioned in this comment (Rosentreter et al. 2018, Lorke et al. 2015, Jeffrey et al. 2018) discuss or quantify the uncertainty associated to the estimate of k via the chamber method but using the water CO2 concentrations obtained via other techniques. In our manuscript we discuss and quantify the uncertainty associated to the estimates of k derived only from CO2 observations inside the chamber. This will be further stressed in the revision of the paper. A fan was not used inside the chamber and the temperature was always almost constant during the chamber incubation. This information will be added in the Method section.

Change:
We have stressed our point by changing the disputed sentence (see L. 79-80) with:
"Furthermore, the uncertainty associated with k estimates relying exclusively on $CO_2$ concentrations gathered using chamber-based CO2 measurements has not been discussed nor quantified by previous studies."

We have specified the chambers have not a fan (see L. 95-96)  and we measured almost constant temperature during the deployments (more than 80% of the runs experienced a variation in temperature less than 2.5 °C) (see L. 148-149).

Comment:
L126 what CO2 sensors? Please add brand, model, and accuracy of CO2 sensor and CO2 analysis.
Answer:
The CO2 sensor specifics are indicated in L. 101-104.
Changes:
We have not made any change here.

Comment:
L127 Roughly, how long did you conduct chamber incubations (runs) for? minutes, half an hour? an hour?
Answer:
Chamber incubation time for steady deployments ranges between 12 mins to 45 mins. This will be specified in the revised methods.
Changes:
We have inserted the range for incubation time of steady deployments (see L. 140-141) and we have explicitly written to refer to Table 2 (see L. 140) for the incubation time of drifting deployments.

Comment:
L134-135 Is this a problem? Even if chamber concentrations inside were not atmospheric, you can still use the change of concentration for estimating k, no? If you measured CO2 every 30sec over the duration of the chamber incubation, then you have a start and end concentration over time (F) that you applied in Eq.2 and Eq.3. Meaning only the difference between start and end concentration is important (slope) and not the concentration itself. I'm curious to hear if the authors agree or disagree.
Answer:
The equilibration condition at the moment of incubation does not affect per se the measure of k. Thus, we agree on the possibility of estimating k also in the case the chamber is not perfectly equilibrated to the atmospheric value at the moment of incubation. In general, ensuing a perfect equilibration before incubation is important to have a realistic representation of the CO2 fluxes occurring at the water-atmosphere interface. Moreover, if the chamber is not equilibrated to the atmosphere you might have a non-homogeneous CO2 concentration inside the chamber at the beginning of the incubation and this could generate a potential bias in the typical exponential curve during the incubation (i.e. Eq. 2 and 3 could be no longer valid). Also, ensuing equilibrium to the atmospheric value before incubation guarantees the maximization of the concentration gradient and this leads to a clearer CO2 signal.
Changes:
We have not made any change here.

Comment:
L139 Were the atmospheric concentrations outside close to 400 ppm?
Answer:
Yes, we will explicitly indicate it around L. 139.
Changes:
We have specified that the sensor showed atmospheric concentration close to 400 ppm (see L. 153).

Comment:
L193 do you mean increasing "linear regression"? If yes, what was your threshold r2?
Answer:
The first data quality check we used was to consider only the curves that showed a monotonous increase (or decrease) in CO2 concentration inside the chamber volume during the incubation process. This was just a visual data quality check. Then we used the NSE coefficient to discriminate the ones with the highest performance. This will be clarified in the revised Methods.
Changes:
We have specified that the first was just a visual test (see L. 206).

Comment:
L250-251 this sentence could be deleted as this is also mentioned in the Table 3 caption.
Answer:
We will delete it. Thank you.
Changes:
We have removed the sentence (see L. 263-265).

Comment:
Figure 4b shouldn't this be k600, not k?
Answer:
No, the figure is referred to a single deployment (there is no need to standardize the k to k600 in this case).
Changes:
We have not made any change here.

Comment:
While this study greatly contributes to our understanding of appropriate chamber design and conditions (drifted vs anchored) of the chamber method in general, I wonder how good this chamber method is in predicting the CO2 flux in comparison to other k methods and empirical k models? For example, were CO2 fluxes measured in the flumes better predicted by k600 derived from the chambers measurements in this study than predicted from k600 models (e.g. Raymond et al. 2013, Ulseth et al. 2020)? Or more practically, would the authors recommend to use FF chamber anchored mode over the k600 model by Ulseth et al. (2020) based on energy dissipation for estimating CO2 fluxes in rivers? Do the empirical models under or overestimate the flux?
Answer:
We thank you for your comment. In this paper, we do not have a direct comparison between the chamber method and other possible k methods. Hence, we are not able to state that the chamber method might be better than other k methods. In general, we observed k estimates from FF chamber in line with empirical models from Raymond et al. 2012 and Ulseth et al. 2019. We are not able to argue that our chamber estimates under or overestimates the flux with respect to these models. Your questions are all valuable, and they provide exciting hints for further research. We are conscious that to have measure of k from other methods might improve the paper and give a further support to the validity of our chamber method. We hope to have the possibility to investigate this further.
Change:
We have no made changes here.

**Additional changes:**

We corrected a typo (see L. 481): The ADV sampling depth was in fact 5-6 cm, not 3.5 cm as originally indicated.

We corrected a typo in L. 518 (m3 should read m).

We added details on the quality of our ADV data to estimate $\varepsilon$. Specifically, we now state in L. 540-541 that 82% of all ADV measurements passed all quality requirements (e.g. SNR > 70, assumption of isotropic turbulence, positive R2 in spectral fitting, etc.) and were used for $\varepsilon$ estimation.

We noticed a mistake in the description of the velocity time series despiking procedure. To identify spikes in the velocity time series, we now use the three-dimensional phase-space method by Goring and Nikora (2002) with modifications by Mori et al. (2007). We did not use the approach originally described that identifies spikes as observations outside a multiple of the standard deviation of neighboring data. We preferred the phase space method because it is not sensitive to subjective threshold levels and window sizes, and because it is well established for despiking ADV data. We interpolated spikes using cubic polynomials, not running medians, following procedures by Mori (2020). We adjusted the methods description (see L. 488-495) and online R code accordingly and provide further references that describe the method in detail.

We changed the order of despiking and coordinate transformation, following recommendations by Doroudian et al. (2010, Limnolgy and Oceanography: Methods 8). We now first despike the data and then transform the coordinate system. We adjusted the methods description and online R code accordingly and moved the sentence in L. 482-483 to L. 493-495.